

# Contrasting determinants for the introduction and establishment success of exotic birds in Taiwan using decision trees models

Shih-Hsiung Liang[1], Bruno Andreas Walther[2] and Bao-Sen Shieh[3,4]

[1] Department of Biotechnology, National Kaohsiung Normal University, Kaohsiung, Taiwan
[2] Master Program in Global Health and Development, College of Public Health, Taipei Medical University, Taipei, Taiwan
[3] Department of Biomedical Science and Environmental Biology, Kaohsiung Medical University, Kaohsiung, Taiwan
[4] Department of Medical Research, Kaohsiung Medical University Hospital, Kaohsiung, Taiwan

## ABSTRACT

**Background**. Biological invasions have become a major threat to biodiversity, and identifying determinants underlying success at different stages of the invasion process is essential for both prevention management and testing ecological theories. To investigate variables associated with different stages of the invasion process in a local region such as Taiwan, potential problems using traditional parametric analyses include too many variables of different data types (nominal, ordinal, and interval) and a relatively small data set with too many missing values.

**Methods**. We therefore used five decision tree models instead and compared their performance. Our dataset contains 283 exotic bird species which were transported to Taiwan; of these 283 species, 95 species escaped to the field successfully (introduction success); of these 95 introduced species, 36 species reproduced in the field of Taiwan successfully (establishment success). For each species, we collected 22 variables associated with human selectivity and species traits which may determine success during the introduction stage and establishment stage. For each decision tree model, we performed three variable treatments: (I) including all 22 variables, (II) excluding nominal variables, and (III) excluding nominal variables and replacing ordinal values with binary ones. Five performance measures were used to compare models, namely, area under the receiver operating characteristic curve (AUROC), specificity, precision, recall, and accuracy.

**Results**. The gradient boosting models performed best overall among the five decision tree models for both introduction and establishment success and across variable treatments. The most important variables for predicting introduction success were the bird family, the number of invaded countries, and variables associated with environmental adaptation, whereas the most important variables for predicting establishment success were the number of invaded countries and variables associated with reproduction.

**Discussion**. Our final optimal models achieved relatively high performance values, and we discuss differences in performance with regard to sample size and variable treatments. Our results showed that, for both the establishment model and introduction model, the number of invaded countries was the most important or second most important determinant, respectively. Therefore, we suggest that future success for

Corresponding author
Bao-Sen Shieh, bsshieh@kmu.edu.tw

introduction and establishment of exotic birds may be gauged by simply looking at previous success in invading other countries. Finally, we found that species traits related to reproduction were more important in establishment models than in introduction models; importantly, these determinants were not averaged but either minimum or maximum values of species traits. Therefore, we suggest that in addition to averaged values, reproductive potential represented by minimum and maximum values of species traits should be considered in invasion studies.

## INTRODUCTION

Biological invasions have become a major threat to biodiversity (*Pimentel, Zuniga & Morrison, 2005*). Hence, some studies of biological invasion have focused on how to prevent the invasion or how to eradicate the invasive species (*Dana, Jeschke & García-de-Lomas, 2014*). As more and more invasive species have spread into the wild, invasive species have also become important subjects in testing ecological theories in relation to niche and competition (e.g., *Broennimann et al., 2007*; *Allen et al., 2015*). Both prevention management and testing ecological theories require the identification of the key factors underlying success at different stages in the invasion process (*Duncan, Blackburn & Sol, 2003*); moreover, factors that are important to explain the invasion success have been suggested to be different at each stage of the invasion process (*Kolar & Lodge, 2002*; *Williamson, 2006*; *Dawson, Burslem & Hulme, 2009*).

Compared with other vertebrate taxa, birds have a higher number of invasive species and invasion success rates in a study focusing on Europe and North America (*Jeschke & Strayer, 2006*). Previous studies on exotic birds have identified two major categories of factors associated with their success at the introduction and establishment stages: human selectivity factors and species traits. Human selectivity factors consist of factors such as taxa and geography selected non-randomly by humans during the transport or introduction stages of exotic birds (*Duncan, Blackburn & Sol, 2003*). Species traits, on the other hand, then play an important role during the introduction and establishment stages (*Blackburn, Cassey & Lockwood, 2009*).

In Taiwan, at least 290 exotic species of pet birds have been imported, and a 9.7% rate of invasion success was estimated (*Shieh et al., 2006*). For the transport stage, non-random selectivity of exotic birds imported to Taiwan was associated with bird family, native geographic range, body size, and song production of species (*Su, Cassey & Blackburn, 2014*); as to the later stages of invasion, pet trade factors such as song attractiveness were significantly associated with introduction success but not establishment success (*Su, Cassey & Blackburn, 2016*).

For the exotic birds of Taiwan, species traits that help to avoid stochastic extinction or to constrain establishment (cf. *Sol, 2008*) have not been investigated with regard to their influences on different stages of the invasion process. To investigate the effects of these

factors which are associated with both human selectivity and species traits onto different stages of the invasion process in a local region such as Taiwan, two potential problems using traditional parametric analyses have been identified as (1) a relatively small data set with too many missing values and (2) too many variables of different types (nominal, ordinal, and interval).

Machine learning is a new, advanced analytical method which overcomes many of the restrictions of traditional parametric analyses. We chose the decision tree method, a machine learning algorithm, because its advantages include no need to input data for missing values and no assumptions about the distribution of the data; therefore, this method is ideal for dealing with mixed data types, such as nominal, ordinal and interval variables (*Olinsky, Kennedy & Brayton Kennedy, 2014*). In studies of biological invasion, the decision tree method was first applied to investigating a data set of 45 fish species for risk assessment in the Great Lakes (*Kolar & Lodge, 2002*). In another recent study, *Chen, Peng & Yang (2015)* found that decision tree methods not only work best with nominal variables but also have higher performance values than traditional parametric methods in predicting alien herb invasion. In a comparative study of trait-based risk assessment for invasive species which included a bird data set, *Keller, Kocev & Džeroski (2011)* found that random forests (an ensemble method that creates multiple decision tree sub-models) was one of the two best performing methods. *Vall-llosera & Sol (2009)* investigated only one of the four stages of the invasion process, namely establishment success, in a global risk assessment for invasive birds. They found that their tree model had an overall predictive accuracy as high as the conventional statistical models (generalized linear mixed models). Besides these two studies, which only focused on the establishment stage for exotic birds using decision tree models, there are, to our knowledge, no other studies which used decision tree methods to analyze the determinants for both the introduction and establishment stages of exotic birds.

Consequently, we decided to use decision tree methods to assess factors associated with human selectivity and species traits which determine the success during the introduction and establishment stages of exotic birds in Taiwan. We used five decision tree models which differed in regard to resampling the data set and compared their performance. An optimal prediction model was chosen based on five performance measures, and the relative importance of factors in the optimal model for introduction success and for establishment success was examined and compared.

## MATERIALS & METHODS

### Species of the data set

The four stages of the invasion process were defined in *Duncan, Blackburn & Sol (2003)* as transport, introduction, establishment, and spread. In this study, we focused on the introduction and establishment stages. For a species to reach the introduction stage, it must have passed the transport stage. Therefore, we selected all the exotic species which had been transported to Taiwan's main island (not including surrounding islands, such as Lanyu and Kinmen Island) as documented in *Shieh et al. (2006)* which included the results of *Chi (1995)*, *Severinghaus (1999)* and *Lin (2004)*. Whether a transported species

has passed the subsequent stages of the invasion process was based (1) on escaping records in the field (introduction success) and (2) breeding record in the field (establishment success). We followed the detailed methods of how to define introduction success and establishment success which were given in *Su, Cassey & Blackburn (2016)*. However, *Su, Cassey & Blackburn (2016)* based their decision of establishment success on the respective species having been recorded to be breeding at least twice; instead, we based it on at least one record of fledglings actually having left the nest successfully.

In order to record all the escaping and breeding records of bird species up to 2015, we continuously (1) checked information from the Chinese Wild Bird Federation (http://www.bird.org.tw/) database which is the main collector of wild bird data in Taiwan, as well as other Taiwanese websites dedicated to natural history observations of birds, (2) remained in contact with local ornithologists, birdwatchers and bird societies, and (3) included any relevant publications (e.g., *Walther, 2011*; *Walther, 2014* for red-whiskered bulbul, *Pycnonotus jocosus*; *Fan et al., 2009* for white-rumped shama, *Copsychus malabaricus*, or *Shieh, Lin & Liang, 2016* for Asian glossy starling, *Aplonis panayensis*). Most of this updated information was published recently in a project report for the Taiwan Forestry Bureau (*Liang & Shieh, 2016*).

Despite following the methods as described in *Su, Cassey & Blackburn (2016)*, we independently collected all the data used in this analysis beginning in 2004 and ending in 2015. Our dataset thus contains 283 full species (although we entered subspecies in our dataset, for this analysis, we only used full species), which were transported to Taiwan (see above). Of these 283 species, 95 species escaped to the field successfully (introduction success). Of these 95 species, 36 species reproduced in the field of Taiwan successfully (establishment success) (see Table S1 for species list).

## Variables

We collected 22 variables for each species, including two nominal ones (order and family), six ordinal ones (latitude overlap with Taiwan: 0–2, migration pattern: 0–3, nesting location: 0–3, feeding: 1–3, diet: 1–6, and habitat: 0–6), three binary ones (hole nest, Taiwan genus_resident, dichromatism), and 11 interval ones (clutch size: clutch, maximum clutch size: Mclutch, incubation days: incubation, minimum incubation days: Minincub, body length: length, maximum body length: Mlength, body mass: Mass, maximum body mass: Mmass, the number of invaded countries: Invcountry_Max, distribution range (km$^2$): Range, the number of subspecies: subspecies) (see supplementary file Table S2 for code descriptions of variables). The variable Taiwan genus_resident was based on the information in *Hsiao & Li (2014)*. For the other variables, we gathered species information from the books of *Del Hoyo et al. (1992–2011)*, *Dunning Jr (1993)*, and internet databases of IUCN (http://www.iucn.org) and BirdLife International Datazone (http://datazone.birdlife.org) (see Table S1 for associated information of each species and Table S2 for code descriptions of variables). When we collected the values for reproduction and body size for each species, we usually found a given range instead of fixed values in the references. In order to account for the maximum adaptation and reproduction potential in the invasion process, we used maximum values such as maximum body mass or minimum

values such as minimum incubation days in addition to averaged values. To determine the number of invaded countries (Invcountry_Max), we counted the total (or maximum) number of countries in which occurrences of introduced populations of each respective species were reported.

## Decision trees models and variable treatments

To investigate the possible effects of nominal variables (family and order) and ordinal variables on the performance of the decision tree models, we conducted three variable treatments for modeling: (I) including all variables, (II) excluding nominal variables, and (III) excluding nominal variables and replacing ordinal values with binary ones; e.g., changing habitat values of 0–4 to 0 (natural habitats) and habitat values of 5–6 to 1 (artificial habitats).

For each variable treatment, we used five decision tree models (DT_no bagging, DT_bagging 90%, DT_bagging 100%, gradient boosting, and HP forest) to predict the outcomes of introduction and establishment, respectively. Modeling processes and comparisons of model performance were implemented using SAS Enterprise Miner 13.1 (for diagrams of process flow, see supplementary files Figs. S1 & S2). Because of the small data set, no data partition was implemented; that is, all data were used as training data. Instead, other methods, such as bagging and cross validation, which have been suggested for the use with small data sets (*SAS Institute Inc., 2013*), were used in the present study.

DT_no bagging is the traditional classification tree method by constructing a layered tree model with the following settings: splitting rule = Gini, cross validation with 10 subsets and 100 repeats. The DT_bagging 90% and DT_bagging 100% methods used the same setting of splitting rule and cross validation as the DT_no bagging method but with bagging 90% or 100% of the data set for 50 times, respectively. Gradient boosting is a boosting method that resamples the data set to produce a series of decision trees which together form a single predictive model which has been found to be less prone to overfitting the data than a single decision tree (*Georges, 2008*). HP Forest is the random forest method which builds many parallel trees forming a forest; a tree in the forest is a sample without replacement from all the available observations, and the input variables that are considered for splitting a node are randomly selected from all the available inputs (*Hall et al., 2014*).

We calculated five performance measures to compare models, namely, the area under the receiver operating characteristic curve (AUROC), the specificity which measures the fraction of negative events that were correctly labeled, the precision which measures the fraction of positively labeled outcomes that were correctly labeled, the recall which measures the fraction of positive events that were correctly labeled, and the accuracy which measures the fraction of all events that were correctly labeled (accuracy = 1 − misclassification rate) (*Söhngen, Chang & Schomburg, 2011*). These five performance measures have the same range (0–1), and we gave each measure equal weight in evaluating the model performance in accordance with *Chen, Peng & Yang (2015)*. The higher the values of these five performance measures are, the better the model performs; therefore, we summed up the five values (from hereupon called the "total score") and chose the model with the

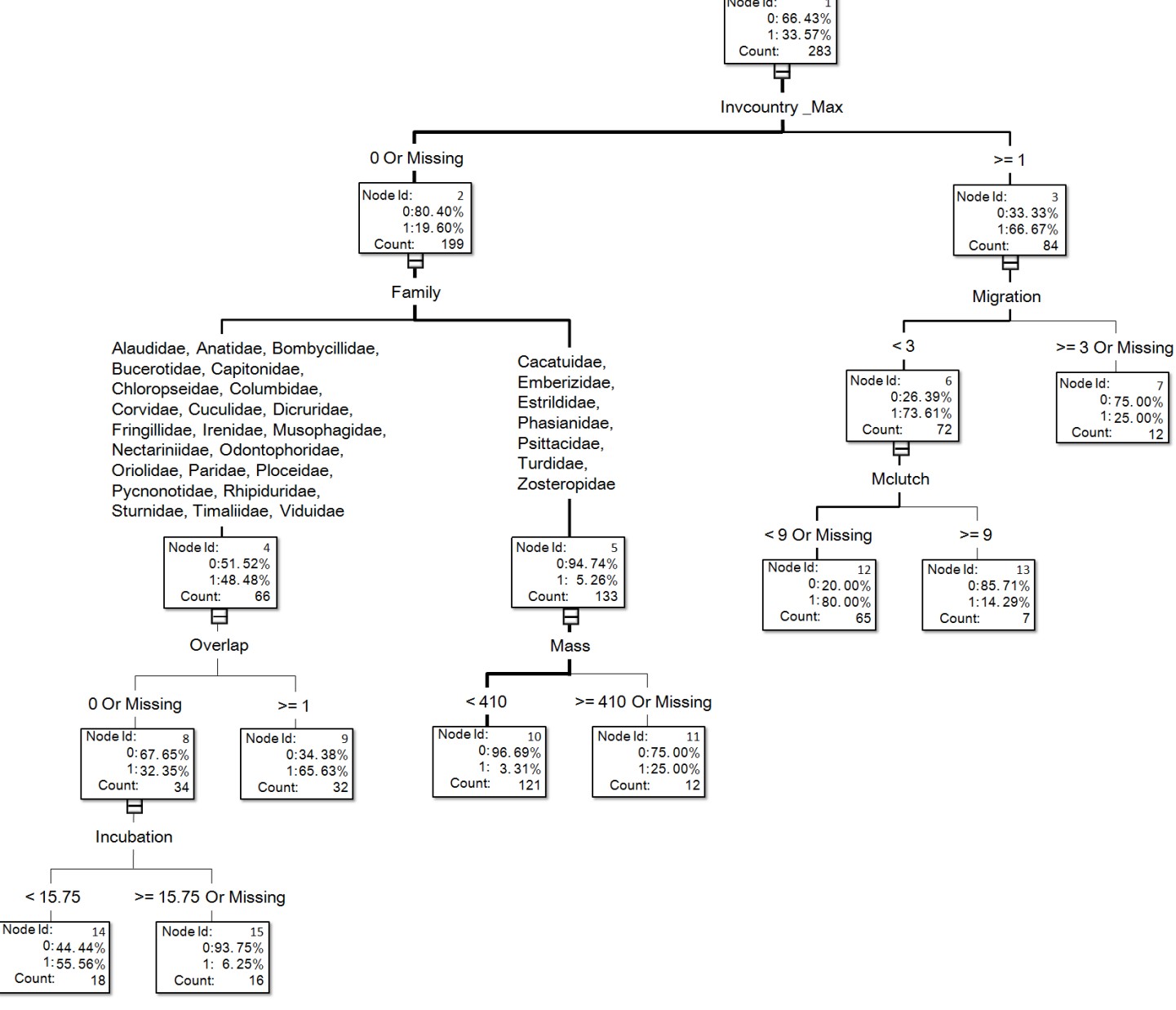

**Figure 1** **The visual output of the introduction model based on the classification tree method for exotic birds of Taiwan generated from the dataset of 283 transported species, of which 95 species successfully escaped in the field (see Table S1 for associated information of each species and Table S2 for code descriptions of variables).**

highest sum as our final optimal model. We then compared the relative importance of each of the variables in the optimal introduction model and establishment model.

For illustrative purposes, we chose the visual output of the resulting trees of DT_no bagging of variable treatment I for our figures (Figs. 1 and 2). Such visual outputs are not possible for the other four methods (namely, DT_bagging 90%, DT_ bagging 100%, gradient boosting, and HP forest).

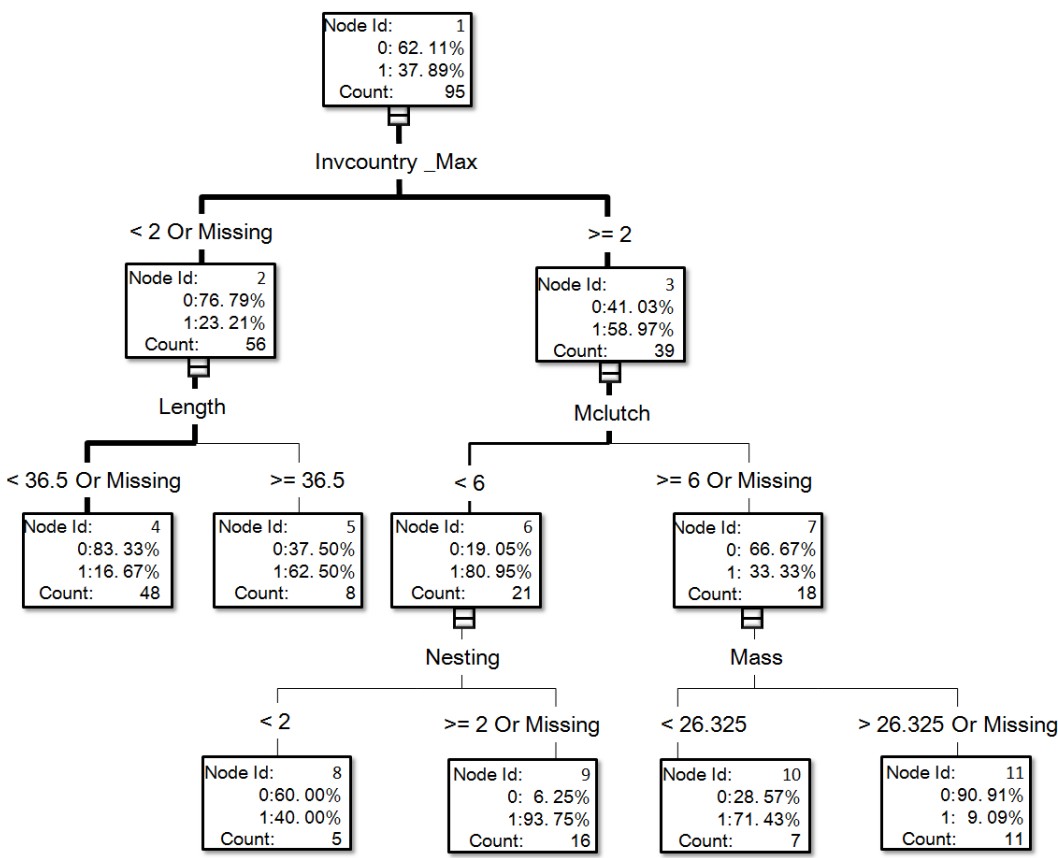

**Figure 2** The visual output of the establishment model based on the classification tree method for exotic birds of Taiwan generated from the dataset of 95 introduced species, of which 36 species successfully reproduced in the field (see **Table S1** for associated information of each species and **Table S2** for code descriptions of variables).

We used the decision tree models described above to build various versions of two kinds of models: (1) introduction success prediction models and (2) establishment success prediction models. However, for brevity's sake, from hereupon we will call them introduction models and establishment models, respectively.

## RESULTS

Across the three variable treatments and for both the introduction models (Table 1) and establishment models (Table 2), the gradient boosting models always achieved the highest score among the five decision tree models (i.e., it performed best overall). However, this overall best performance did not mean that gradient boosting always performed best when comparing values of the five performance measures. For instance, Table 1 (see also supplementary file Fig. S3 for receiver operating characteristic curves, and supplementary file Fig. S4 for classification charts) shows that gradient boosting only performed best for accuracy in variable treatment I and II; otherwise, other models always performed better using the other four performance measures. Nevertheless, across all three treatments, the total score is always highest for gradient boosting for the introduction models (Table 1).

**Table 1** Comparison of five performance measures among five introduction models of exotic birds in Taiwan, separately for three variable treatments (see 'Methods' for details).

| Model | AUROC | Specificity | Precision | Recall | Accuracy | Total |
|---|---|---|---|---|---|---|
| **Variable treatment I** | | | | | | |
| DT_no bagging | 0.894 | 0.830 | 0.722 | 0.874 | 0.845 | 4.164 |
| DT_bagging 90% | 0.970 | 0.936 | 0.782 | 0.453 | 0.774 | 3.914 |
| DT_bagging 100% | 0.976 | 0.910 | 0.742 | 0.516 | 0.777 | 3.921 |
| Gradient boosting | 0.936 | 0.941 | 0.869 | 0.768 | 0.883 | 4.398 |
| HP Forest | 0.903 | 0.963 | 0.873 | 0.505 | 0.809 | 4.053 |
| **Variable treatment II** | | | | | | |
| DT_no bagging | 0.904 | 0.872 | 0.765 | 0.821 | 0.855 | 4.217 |
| DT_bagging 90% | 0.949 | 0.899 | 0.683 | 0.432 | 0.742 | 3.705 |
| DT_bagging 100% | 0.955 | 0.910 | 0.742 | 0.516 | 0.777 | 3.900 |
| Gradient Boosting | 0.924 | 0.915 | 0.816 | 0.747 | 0.859 | 4.261 |
| HP Forest | 0.894 | 0.963 | 0.848 | 0.411 | 0.777 | 3.893 |
| **Variable treatment III** | | | | | | |
| DT_no bagging | 0.910 | 0.888 | 0.781 | 0.789 | 0.855 | 4.224 |
| DT_bagging 90% | 0.946 | 0.910 | 0.691 | 0.400 | 0.739 | 3.685 |
| DT_bagging 100% | 0.953 | 0.888 | 0.700 | 0.516 | 0.763 | 3.820 |
| Gradient Boosting | 0.919 | 0.926 | 0.827 | 0.705 | 0.852 | 4.229 |
| HP Forest | 0.888 | 0.957 | 0.840 | 0.442 | 0.784 | 3.912 |

**Table 2** Comparison of five performance measures among five establishment models of exotic birds in Taiwan, separately for three variable treatments (see 'Methods' for details).

| Model | AUROC | Specificity | Precision | Recall | Accuracy | Total |
|---|---|---|---|---|---|---|
| **Variable treatment I** | | | | | | |
| DT_no bagging | 0.839 | 0.898 | 0.806 | 0.694 | 0.821 | 4.059 |
| DT_bagging 90% | 0.945 | 0.932 | 0.800 | 0.444 | 0.747 | 3.869 |
| DT_bagging 100% | 0.963 | 0.949 | 0.842 | 0.444 | 0.758 | 3.957 |
| Gradient Boosting | 0.985 | 1.000 | 1.000 | 0.861 | 0.947 | 4.793 |
| HP Forest | 0.901 | 0.983 | 0.875 | 0.194 | 0.684 | 3.638 |
| **Variable treatment II** | | | | | | |
| DT_no bagging | 0.839 | 0.898 | 0.806 | 0.694 | 0.821 | 4.059 |
| DT_bagging 90% | 0.942 | 0.932 | 0.800 | 0.444 | 0.747 | 3.866 |
| DT_bagging 100% | 0.963 | 0.949 | 0.842 | 0.444 | 0.758 | 3.957 |
| Gradient boosting | 0.976 | 0.983 | 0.969 | 0.861 | 0.937 | 4.726 |
| HP Forest | 0.914 | 1.000 | 1.000 | 0.167 | 0.684 | 3.765 |
| **Variable treatment III** | | | | | | |
| DT_no bagging | 0.839 | 0.898 | 0.806 | 0.694 | 0.821 | 4.059 |
| DT_bagging 90% | 0.936 | 0.932 | 0.800 | 0.444 | 0.747 | 3.860 |
| DT_bagging 100% | 0.940 | 0.949 | 0.842 | 0.444 | 0.758 | 3.934 |
| Gradient boosting | 0.971 | 1.000 | 1.000 | 0.778 | 0.916 | 4.665 |
| HP Forest | 0.912 | 1.000 | 1.000 | 0.139 | 0.674 | 3.725 |

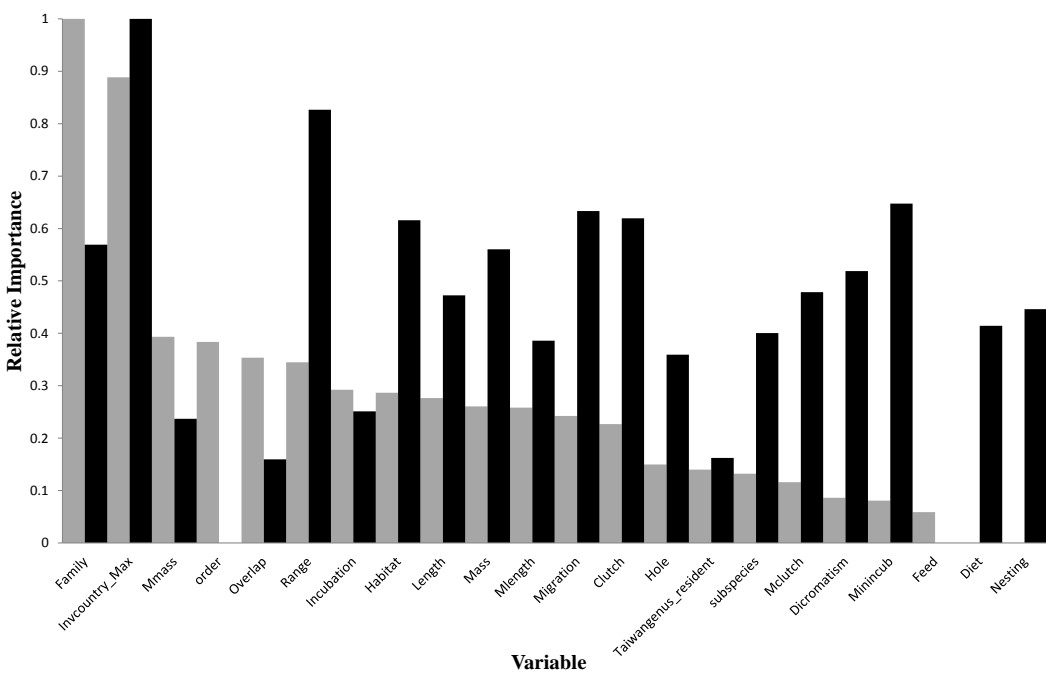

**Figure 3** **Relative importance of variables in the prediction models using the gradient boosting approach (grey bars for introduction models and black bars for establishment models).** For descriptions of codes for variables, see Table S2.

For the establishment models (Table 2, see also supplementary file Fig. S5 for receiver operating characteristic curves, and supplementary file Fig. S6 for classification charts), however, gradient boosting has the highest total score for all the three treatments and also for most of the five performance measures (the only exceptions being specificity and precision in variable treatment II). Therefore, we considered gradient boosting the optimal model for both the introduction models and establishment models and only considered its results from hereupon.

Looking across the three different variable treatment methods I–III, gradient boosting performed best with variable treatment I for the introduction models (Table 1) as well as the establishment models (Table 2). For variable treatments II and III, the total score decreased by only 0.169 (4%) and 0.128 (3%), respectively. We also note that this decreasing trend across variable treatments is maintained for most of the five performance measures. Furthermore, the values of the performance measures are all >0.7 and 60% are >0.9, which means that the performance was consistently high or very high.

In the optimal introduction model, family and the number of invaded countries (Invcountry_Max) were the most important variables, and their relative importance values were 1 and 0.888, respectively (Fig. 3). The top six variables with an importance value >0.3 also included maximum body mass (0.394), order (0.384), latitude overlap with Taiwan (0.354), and distribution range (0.345). For the introduction model based on the classification tree method (Fig. 1), the number of invaded countries was the most important determinant, as it appeared at the top of the tree, which means that the 84 species with

any record of invading other countries had a 66.7% chance of successful introduction to Taiwan. Among these 84 species, the 72 species which had a migration pattern categorized as sedentary (0), local movement (1) or partial migration (2) had a 73.6% chance of successful introduction, while the 12 species categorized as migrants (3) had only a 25.0% chance of successful introduction. Among the 199 species which had no record of invading other countries, family was chosen as an important determinant of successful introduction.

In the optimal establishment model, the number of invaded countries and distribution range were the most important variables, and their relative importance values were 1 and 0.826, respectively (Fig. 3). The top six variables with an importance value >0.6 also included minimum incubation days (Minincub, 0.647), migration pattern (Migration, 0.633), clutch size (Clutch, 0.62), and habitat type (Habitat, 0.616). The relative importance of the variable family decreased to 0.569 which is therefore much lower than in the optimal introduction model (see above). For the establishment model based on the classification tree method (Fig. 2), the number of invaded countries was again the most important determinant, as it appeared at the top of the tree. In this case it means that the 39 species with a record of invading at least two countries had a 59.0% chance of successful establishment in Taiwan, while the 56 species with a record of invading fewer than two countries had only a 23.2% chance of successful establishment. Among the 39 species noted above, the 21 species with a maximum clutch size (Mclutch) <5.5 had an 81.0% chance of successful establishment, while the other 18 with a maximum clutch size of ≥5.5 had only a 33.3% chance of successful establishment. Finally, among the 56 species noted above, the eight species with a body length (Length) ≥36.5 cm had a 62.5% chance of successful establishment.

## DISCUSSION

### Model comparisons and variable treatment comparisons

Our results showed that for the complete data set of 283 transported species or for the data set of 95 introduced species, the gradient boosting method performed better than the other four decision tree methods. While we calculated five performance measures, the only other study which used the decision tree method on a bird data set was *Keller, Kocev & Džeroski (2011)* who calculated only AUROC and accuracy values. Considering AUROC values first, the AUROC values of gradient boosting of our study were over 0.919 in the introduction models and over 0.971 in the establishment models; thus, they were all higher than our values for the random forests method. This is in contrast to the results of *Keller, Kocev & Džeroski (2011)* who found that, based on the AUROC values, random forests performed better than gradient boosting for both their New Zealand and Australia bird data sets. Specifically, AUROC values for gradient boosting for their New Zealand (79 species with 11 traits) and Australia (52 species with 11 traits) data sets were 0.682 and 0.681, respectively, whereas AUROC values for random forests were 0.731 and 0.745, respectively. *Pearce & Ferrier (2000)* suggested that AUROC values between 0.7 and 0.9 indicate a reasonable discrimination ability of models, and values higher than 0.9 indicate a very good discrimination ability of models. The higher AUROC values of our study might

have resulted from the inclusion of more variables (up to 22 variables) rather than larger samples used for analysis. In our study, both the introduction model and establishment model used 22 variables, and we found higher AUROC values (0.971–0.985) in the smaller data set (namely, the establishment model with 95 species) than in the larger data set (namely, the introduction model with 283 species) (AUROC values 0.919–0.936). We therefore suggest that even a small data set (less than 100 species) with up to 22 variables can achieve a prediction model of good performance using the gradient boosting method.

Comparing the performances of variable treatment I with variable treatments II and III, we found little difference on model performance. Treatment II excluded nominal variables, and treatment III changed ordinal variables of species traits into binary variables, but neither one of these changes really had much discernable influence on overall performance. Our results therefore provide evidence to support the use of ordinal variables of species traits, and that there is no need to convert ordinal variables of species traits to binary ones for their use in decision tree models.

## Predictors of introduction and establishment success in exotic birds

Perhaps the most interesting and novel result of our study is that, for both the establishment model and introduction model, the number of previously invaded countries was the most important or second most important determinant in all the models. Therefore, our study suggests that future success for introduction and establishment of birds can be gauged by simply looking at previous success in invading other countries or regions. Future studies should include this variable to confirm our supposition because it might be a very simple and straightforward way to predict the potential invasion success of a species: if it has been successful before, it will probably be successful again. While this variable could not have been established a few decades ago, we now have a global track record of successful species invasions, and we might therefore be able to use it to better predict future local or regional invasions. Furthermore, global studies could investigate what species traits and other relevant factors, e.g., local ecological factors, are related to the number of successfully invaded countries; or, given the differential size of countries, the actual area invaded.

Another important determinant was family. While family was the most important variable in the optimal introduction model, it dropped to being only the seventh most important variable in the optimal establishment model. In other words, family was an important determinant of introduction but not establishment in Taiwan. Our results thus differ from those of a global study which found that bird family was also a good predictor for establishment success (*Lockwood, 1999*). The discrepancy between this study and our study could result from the fact that exotic birds in Taiwan are primarily introduced for aesthetic reasons but not for hunting (*Shieh et al., 2006*; *Su, Cassey & Blackburn, 2016*), while the global data set included many hunted species.

Several species traits were also chosen as important determinants for the introduction and establishment models. For the optimal introduction model, the top three selected species traits were maximum body mass (Mmass), latitude overlap with Taiwan (Overlap), and distribution range (Range). Among these three variables, maximum body mass was ranked the most important, and it also had a relative importance greater than that of two

other closely related measures, specifically, the averaged body mass (Mass) and body length (Length). One possible explanation is that birds are usually heavier in captivity under well fed condition. Our data set contained primarily pet species (and not game species, which are prevalent in many other studies), and the body mass of pet birds might be higher than the average body mass of their wild congeners and therefore closer to the maximum attainable body mass. In order to consider the representability and the maximum adaptation potential in the invasion process, we therefore suggest that including maximum body mass may be important in order not to miss a potentially important determinant for the invasion success of exotic pet birds in particular. For example, *Su, Cassey & Blackburn (2016)* did not find that body mass had any influence on introduction success. However, they only used averaged body mass, and perhaps their result would have been different if they had also included maximum body mass. Furthermore, *Cassey*'s (*2001*) global study found that averaged body mass was significantly correlated with introduction success which further supports the role of some measure of body mass being an important determinant of introduction success.

Finally, several species traits related to reproduction were also important, such as minimum incubation days (Minincub), clutch size (Clutch), dichromatism, and nesting location (Nesting); however, these determinants were more important in establishment success than in introduction success. Furthermore, given that some top ranking variables were associated with maximum or minimum values of species traits, we suggest that in addition to averaged values, reproductive potential represented by minimum and maximum values of species traits should be considered in prediction models of invasion studies.

We conclude that decision tree models are efficient for the analysis of small data sets with mixed types of variables, including nominal, ordinal and interval variables, in predicting the invasion success of exotic birds. Our results further demonstrate that the most important determinants in predicting introduction success of exotic birds in Taiwan were the bird family, the number of invaded countries, and variables associated with environmental adaptation, whereas the most important determinants in predicting establishment success were the number of invaded countries and variables associated with reproduction.

## ACKNOWLEDGEMENTS

We thank Liviu Parau and two anonymous reviewers for their valuable comments and suggestions on previous drafts of the manuscript.

### Funding

This work was supported by the Ministry of Science and Technology, Taiwan, R.O.C. (grant no. MOST 105 - 2311 - B - 037 – 002) and the Forestry Bureau of the Taiwanese Government. The funders had no role in study design, data collection and analysis, decision to publish, or preparation of the manuscript.

## Grant Disclosures

The following grant information was disclosed by the authors:
Ministry of Science and Technology, Taiwan, R.O.C.: MOST 105 - 2311 - B - 037 – 002.
Forestry Bureau of the Taiwanese Government.

## Competing Interests

The authors declare there are no competing interests.

## Author Contributions

- Shih-Hsiung Liang and Bao-Sen Shieh conceived and designed the experiments, performed the experiments, analyzed the data, contributed reagents/materials/analysis tools, wrote the paper, prepared figures and/or tables, reviewed drafts of the paper.
- Bruno Andreas Walther conceived and designed the experiments, contributed reagents/materials/analysis tools, wrote the paper, prepared figures and/or tables, reviewed drafts of the paper.

## Data Availability

    The raw data has been supplied as a Supplementary File.

## Supplemental Information

Supplemental information for this article can be found online at http://dx.doi.org/10.7717/peerj.3092#supplemental-information.

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
