# Peer review of "Contrasting determinants for the introduction and establishment success of exotic birds in Taiwan using decision trees models"

_PeerJ, doi:10.7717/peerj.3092_

## Round 0.1 · original submission · Major Revisions

· Academic Editor

Major Revisions

Dear authors

Thank you for submitting your work to our journal. As you can see our reviewers would like to see some improvements of your ms.

I hope to get your revision in due time

Kind regards,

Michael Wink
Academic editor

Reviewer 1 ·

Basic reporting

Figs. 2 and 3 need manual editing, since floating-number thresholds do not make sense for integer variables.

Experimental design

There is a conflict between the sentence in lines 101 and 102 and the sentence starting in line 269. It needs to clarified how often this approach had been used before in the same field and in the wider area.
The sentence in lines 162-164 needs to be reformulated as it appears as if countries with native ("feral") populations were counted for "number of invaded countries".

Validity of the findings

Even if the number of invaded countries is correctly defined (see above), the question remains what it tells us biologically that this variable turned out to be the most relevant (and practical). What drove species to have already invaded so many countries? One should re-analyze the data with "number of invaded countries" as response variable.

Comments for the author

It is a great achievement to apply this method to this question and dataset. In lines 239-263, the output of decision trees is clearly illustrated.

Minor issues:
l. 28: introduction success
l. 122: replace "to establish"
from l. 146: do not use a tilde for a range as it is commonly used for model descriptions (e.g. in R)
l. 250: Fig. 3

·

Basic reporting

No comment.

Experimental design

No comment.

Validity of the findings

No comment.

Comments for the author

My recommendation is to include one more variable: price (monetary value) of 1 individual from each respective bird species. I anticipate it would be difficult to obtain an accurate economical value, and therefore suggest 4 price categories: <50 euro/ US dollars, 50-100, 100-500 and >500.
Although I assume this addition will not change the results significantly, I believe it will help to better explain the main findings.

Line 151 (or 7th line in the “Variables” paragraph) Invcountry_Max(now is 'contry'). I found the same mistake in S1, species list, column Z.
Line 245-246 which had no record / with no record: a word is missing.

Reviewer 3 ·

Basic reporting

The reporting is clear and well written. The concise introduction contains all the information necessary. Overall, the paper complies to the professional standards to be expected for a scientific report.

Experimental design

The statistical methods were described precisely and sufficiently detailed. The decision tree method is not only an interesting alternative to other statistical procedures, it also provides clear rules for practitioners. The results obtained shed light on some important issues in invasion biology and hence conservation.

Validity of the findings

The findings are based on a well researched data set and are convincing. The discussion does include statements on the discrepancies with previous results by other research groups.

Comments for the author

I would like to see a statement on the justification of using a simple sum of (highly non-linear) performance measures for assessing the overall performance of a method. The authors may also want to speculate a bit on the finding that maximum body size was a better predictor than average body size. To me this looks a bit a statistical fluke, possibly caused by one or very few outliers.

---

## Round 0.2 · Minor Revisions

· Academic Editor

Minor Revisions

Dear authors

You are almost through the procedure; some small attention to Figs S1 and D2 is still needed

Regards
Michael Wink

Reviewer 1 ·

Basic reporting

no comment

Experimental design

no comment

Validity of the findings

no comment

Comments for the author

I see all issues raised in the first round of reviewing sufficiently addressed.

In Figs. S1 and S2, the resolution is too low to read anything properly. This needs to be adjusted, e.g. by providing vector graphic files.

---

## Round 0.3 · accepted · Accept

· Academic Editor

Accept

Dear authors,

Congratulations - your ms is accepted

Regards,

Michael Wink
Academic editor